# Hour-1 bundle adherence was associated with reduction of in-hospital mortality among patients with sepsis in Japan

Yutaka Umemura[1,2], Toshikazu Abe[3,4,5]*, Hiroshi Ogura[2], Seitato Fujishima[6], Shigeki Kushimoto[7], Atsushi Shiraishi[8], Daizoh Saitoh[9], Toshihiko Mayumi[10], Yasuhiro Otomo[11], Toru Hifumi[12], Akiyoshi Hagiwara[13], Kiyotsugu Takuma[14], Kazuma Yamakawa[15], Yasukazu Shiino[16], Taka-aki Nakada[17], Takehiko Tarui[18], Kohji Okamoto[19], Joji Kotani[20], Yuichiro Sakamoto[21], Junichi Sasaki[22], Shin-ichiro Shiraishi[23], Ryosuke Tsuruta[24], Tomohiko Masuno[25], Naoshi Takeyama[26], Norio Yamashita[27], Hiroto Ikeda[28], Masashi Ueyama[29,30], Satoshi Gando[31,32]

1 Division of Trauma and Surgical Critical Care, Osaka General Medical Center, Osaka, Japan, 2 Department of Traumatology and Acute Critical Medicine, Osaka University Graduate School of Medicine, Osaka, Japan, 3 Department of Emergency and Critical Care Medicine, Tsukuba Memorial Hospital, Tsukuba, Japan, 4 Health Services Research and Development Center, University of Tsukuba, Tsukuba, Japan, 5 Department of Health Services Research, Faculty of Medicine, University of Tsukuba, Japan, 6 Center for General Medicine Education, Keio University School of Medicine, Tokyo, Japan, 7 Division of Emergency and Critical Care Medicine, Tohoku University Graduate School of Medicine, Sendai, Japan, 8 Emergency and Trauma Center, Kameda Medical Center, Chiba, Japan, 9 Division of Traumatology, Research Institute, National Defense Medical College, Tokorozawa, Japan, 10 Department of Emergency Medicine, School of Medicine, University of Occupational and Environmental Health, Kitakyushu, Japan, 11 Trauma and Acute Critical Care Center, Medical Hospital, Tokyo Medical and Dental University, Tokyo, Japan, 12 Department of Emergency and Critical Care Medicine, St. Luke's International Hospital, Tokyo, Japan, 13 Center Hospital of the National Center for Global Health and Medicine, Tokyo, Japan, 14 Emergency & Critical Care Center, Kawasaki Municipal Kawasaki Hospital, Kawasaki, Japan, 15 Department of Emergency Medicine, Osaka Medical and Pharmaceutical University, Osaka, Japan, 16 Department of Acute Medicine, Kawasaki Medical School, Okayama, Japan, 17 Department of Emergency and Critical Care Medicine, Chiba University Graduate School of Medicine, Chiba, Japan, 18 Department of Emergency Medical Care, Kyorin University Faculty of Health Sciences, Tokyo, Japan, 19 Department of Surgery, Center for Gastroenterology and Liver Disease, Kitakyushu City Yahata Hospital, Kitakyushu, Japan, 20 Division of Disaster and Emergency Medicine, Department of Surgery Related, Kobe University Graduate School of Medicine, Kobe, Japan, 21 Emergency and Critical Care Medicine, Saga University Hospital, Saga, Japan, 22 Department of Emergency and Critical Care Medicine, Keio University School of Medicine, Tokyo, Japan, 23 Department of Emergency and Critical Care Medicine, Aizu Chuo Hospital, Fukushima, Japan, 24 Advanced Medical Emergency & Critical Care Center, Yamaguchi University Hospital, Yamaguchi, Japan, 25 Department of Emergency and Critical Care Medicine, Nippon Medical School, Tokyo, Japan, 26 Advanced Critical Care Center, Aichi Medical University Hospital, Aichi, Japan, 27 Advanced Emergency Medical Service Center Kurume University Hospital, Kurume, Japan, 28 Department of Emergency Medicine, Teikyo University School of Medicine, Tokyo, Japan, 29 Department of Trauma, Critical Care Medicine, and Burn Center, Japan Community Healthcare Organization, Chukyo Hospital, Nagoya, Japan, 30 Community Healthcare Organization, Chukyo Hospital, Nagoya, Japan, 31 Department of Anesthesiology and Critical Medicine, Hokkaido University Faculty of Medicine, Sapporo, Japan, 32 Department of Acute and Critical Care Medicine, Sapporo Tokushukai Hospital, Sapporo, Japan

* abetoshi111@gmail.com

## Abstract

### Background

The updated Surviving Sepsis Campaign guidelines recommend a 1-hour window for completion of a sepsis care bundle; however, the effectiveness of the hour-1 bundle has not

the corresponding author upon reasonable request through the Japanese Association for Acute Medicine for researchers who meet the criteria for access to confidential data. Alternatively, data are available from the Japanese Association for Acute Medicine Ethics Committee with the following contact information: e-mail: jaam-6@bz04.plala.or.jp, name of dataset: JAAM MAESTRO.

**Funding:** The funders had no role in study design, data collection and analysis, decision to publish, or preparation of the manuscript.

**Competing interests:** The authors have declared that no competing interests exist.

**Abbreviations:** SSC, Surviving Sepsis Campaign; JAAM, Japanese Association for Acute Medicine; MAESTRO, Multicenter Assessment for Sepsis Treatment and Outcome; ICU, intensive care units; SOFA, Sequential Organ Failure Assessment; CCI, Charlson comorbidity index; EGDT, early goal-directed therapy.

been fully evaluated. The present study aimed to evaluate the impact of hour-1 bundle completion on clinical outcomes in sepsis patients.

## Methods

This was a multicenter, prospective, observational study conducted in 17 intensive care units in tertiary hospitals in Japan. We included all adult patients who were diagnosed as having sepsis by Sepsis-3 and admitted to intensive care units from July 2019 to August 2020. Impacts of hour-1 bundle adherence and delay of adherence on risk-adjusted in-hospital mortality were estimated by multivariable logistic regression analyses.

## Results

The final study cohort included 178 patients with sepsis. Among them, 89 received bundle-adherent care. Completion rates of each component (measure lactate level, obtain blood cultures, administer broad-spectrum antibiotics, administer crystalloid, apply vasopressors) within 1 hour were 98.9%, 86.2%, 51.1%, 94.9%, and 69.1%, respectively. Completion rate of all components within 1 hour was 50%. In-hospital mortality was 18.0% in the patients with and 30.3% in the patients without bundle-adherent care (p = 0.054). The adjusted odds ratio of non-bundle-adherent versus bundle-adherent care for in-hospital mortality was 2.32 (95% CI 1.09–4.95) using propensity scoring. Non-adherence to obtaining blood cultures and administering broad-spectrum antibiotics within 1 hour was related to in-hospital mortality (2.65 [95% CI 1.25–5.62] and 4.81 [95% CI 1.38–16.72], respectively). The adjusted odds ratio for 1-hour delay in achieving hour-1 bundle components for in-hospital mortality was 1.28 (95% CI 1.04–1.57) by logistic regression analysis.

## Conclusion

Completion of the hour-1 bundle was associated with lower in-hospital mortality. Obtaining blood cultures and administering antibiotics within 1 hour may have been the components most contributing to decreased in-hospital mortality.

## Introduction

Sepsis is now defined "as life-threatening organ dysfunction caused by a dysregulated host response to infection" [1]. Although tremendous progress in medical management has been made over the past few decades, sepsis remains a leading cause of death and enormously impacts global health systems, with approximately 11 million sepsis-related deaths reported in 2017 [2].

Multidisciplinary intensive care has been mainly required as sepsis is a fatal disease, and no specific medicines have been developed to resolve it. The care bundle has recently become a key factor in multidisciplinary intensive care. In the Surviving Sepsis Campaign (SSC) guidelines, to improve awareness and outcomes of sepsis, time-dependent bundles such as time to antibiotic administration have become one of the key components [3–6]. The updated 2018 SSC guidelines recommend a 1-hour window for completion of a sepsis care bundle following the recognition of sepsis to be a reasonable approach instead of within 3 hours from triage or

recognition of sepsis [7]. This update has been intensively debated and remains controversial [8].

The effectiveness of the hour-1 bundle has not been fully evaluated because fundamentally, initiation of earlier treatment does not negatively affect the disease. Several recent studies showed significant associations between bundle-adherent care completed within 3 hours and lower mortality. To date, however, little evidence has been provided of an association between completion of the hour-1 bundle components within 1 hour and lower mortality [9, 10]. In addition, a randomized controlled trial in the pre-hospital setting that assessed early antibiotic use in patients with suspected infection showed that it failed to reduce mortality [11]. Herein, we hypothesized that conventional bundle-adherent care within 3 hours would not be sufficiently effective to reduce the mortality of sepsis and septic shock, and adherence to the hour-1 bundle could play an important role in improving outcomes. Therefore, our aim was to prospectively evaluate adherence to the hour-1 care bundle. We also evaluated the association between completion of the hour-1 bundle and patient outcomes.

## Materials and methods

### Ethics approval and consent to participate

The study protocol was reviewed and approved by the ethics committee of all participating institutions in the Japanese Association for Acute Medicine (JAAM) study group. Osaka University, the representative for the JAAM Multicenter Assessment for Sepsis Treatment and Outcome (MAESTRO) study, was responsible for the overall approval (IRB number 18323). The board waived the requirement for informed consent because of the anonymous nature of the data and because no information on individual patients, hospitals, or treating physicians was obtained.

### Design and setting

This multicenter, prospective, observational study, JAAM MAESTRO (UMIN000036349), was conducted in 17 intensive care units (ICUs) in Japan from July 2019 to August 2020.

### Participants

Patients were eligible for this study if they met the following criteria: 1) were older than 16 years; 2) fulfilled the Sepsis-3 criteria [6], i.e., had a proven or suspected infection and an acute increase of 2 or more points in the Sequential (Sepsis-Related) Organ Failure Assessment (SOFA) score; and 3) were diagnosed as having only new-onset infection.

Exclusion criteria included 1) patients with cardiopulmonary arrest on hospital arrival; 2) patients with the limitation of sustained life care or post-cardiopulmonary arrest resuscitation status at the time of sepsis diagnosis; 3) patients deemed ineligible as study participants by a research director; and 4) patients transferred from other hospitals.

### Data collection

Data were extracted from the MAESTRO database and compiled by the MAESTRO investigators. Collected variables included relevant patient information such as demographics, comorbidities, vital signs, laboratory data, and site of infection. We also obtained data on adherence to sepsis care bundles (specifically, the hour-1 bundle).

In-hospital mortality was identified as the primary outcome. Secondary outcomes were the number of ventilator-free days and ICU-free days, length of hospital stay, and condition at discharge.

Data collection was conducted as part of the clinical routine workup. The MAESTRO site investigators recorded all data throughout the patient's hospital stay. In the case of missing data, the MAESTRO committee requested a reconfirmation of data extraction from the MAESTRO investigators.

## Data definitions

Sepsis care bundles were defined according to SSC guidelines [12] as whether all bundle components were achieved within the appropriate time frame (i.e., 1 hour) and whether they adhered to the indications. Thus, if a component of the bundle was not applicable, we treated achievement of the other components as completion of the bundle (i.e., in cases where administration of crystalloid and application of vasopressor were not indicated), and adherence was defined when the other three components were completed. For all patients, bundle initiation time was defined as the time of sepsis recognition in the emergency department, ward, or ICU. Sepsis recognition was based on clinical judgement, by which the physician-in-charge suspected sepsis at the initial evaluation. The timestamp was recorded in the database by the physician-in-charge.

## Analysis

In total, 100 participants were required in the present study for the bundle-adherent group to conduct multivariate regression analyses. In a previous study conducted in Japan, early administration of antibiotics (within 1 hour) was adhered to in approximately 33% of septic patients [13]. Therefore, a total sample size of 300 patients was assumed to be necessary to include 100 patients with bundle-adherent care.

We divided the patients into two groups, those receiving bundle-adherent care (bundle-adherent group) and those not receiving it (non-bundle adherent group). We performed univariate analyses of the characteristics of the patients in whom the hour-1 bundle was or was not completed within 1 hour. Continuous data are expressed as mean (SD) or median (interquartile range), depending on normality. Categorical variables are shown as proportions. We also evaluated the time to completion of each component of the hour-1 bundle.

The impact of non-adherence to the hour-1 bundle on risk-adjusted in-hospital mortality was estimated by logistic regression analyses adjusted by an inverse probability of treatment weighting analysis using propensity scoring. The propensity score for adherence to the hour-1 bundle was determined using a logistic regression with the following covariates as independent variables, which were specified *a priori* based on clinical experience and prior studies: patient age, sex, admission source (emergency department, ward, or in ICU), Charlson comorbidity index (CCI), mechanical ventilation use, and each organ score within the SOFA (S1 Table). In addition, after replacing time to completion of each component of the hour-1 bundle as a continuous variable, we performed a multivariable logistic regression analysis adjusted for clinically plausible and relevant confounders equal to the covariates to calculate the propensity score. We also fitted logistic regression models to evaluate the association between the increase in mortality and failure or delay in achieving the hour-1 bundle in subgroups with and without septic shock by including product terms between achieving the bundle and the presence of shock. No assumptions were made about these data because the amount of missing data was low.

## Sensitivity analysis

Because two components (measure initial lactate level and begin rapid administration of crystalloid) in the hour-1 bundle were completed in almost all of the patients, we performed the

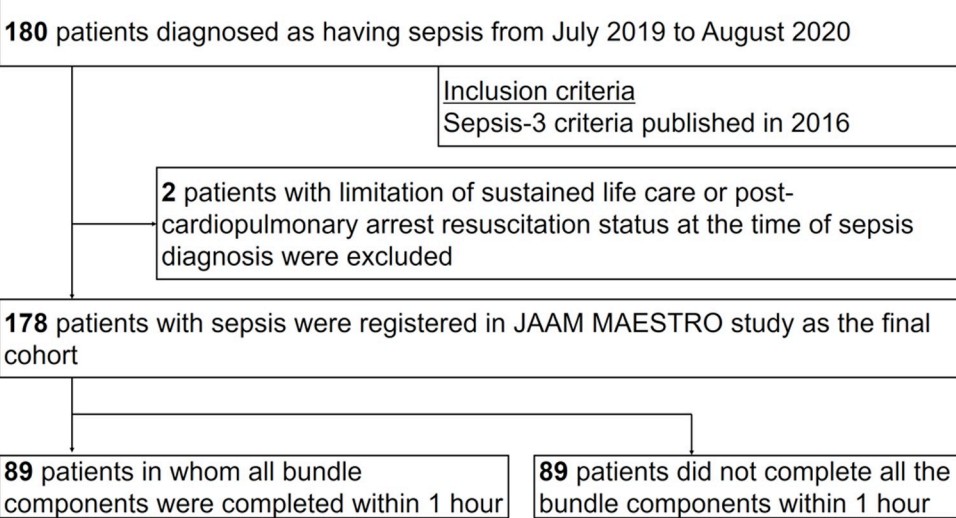

**Fig 1. Patient flow diagram.** *JAAM MAESTRO* Japanese Association for Acute Medicine Multicenter Assessment for Sepsis Treatment and Outcome.

same analyses excluding these two components. Two-tailed p values < 0.05 were considered to indicate significance. All statistical analyses were performed using STATA software version 15.0 (Stata Corp, College Station, TX, USA).

## Results

Among the 180 patients, 178 individuals who met all eligibility criteria entered the final analyses. Of them, 89 patients received bundle-adherent care (Fig 1).

Baseline characteristics, vital signs, laboratory test results, and severity scores obtained at the time of sepsis recognition were similar between the two groups, with the exception of body temperature, white blood cell count, and mechanical ventilation use (Table 1).

We show time to completion of each component of the hour-1 bundle in S2 Table. Two components (measure initial lactate level [98.9%] and begin rapid administration of crystalloid [94.9%]) were completed in 1 hour in almost all patients. The rate of completion of all components within 1 hour was 50%. In-hospital mortality was 18.0% in the bundle-adherent group and 30.3% in the non-bundle-adherent group (p = 0.054) (Table 2). The number of ventilator-free days and ICU-free days and length of hospital stay were not different between the two groups.

The adjusted odds ratio (OR) of the non-bundle-adherent group versus bundle-adherent group for in-hospital mortality was 2.32 (95% CI 1.09–4.95) using an inverse probability of treatment weighting analysis with propensity score (Fig 2).

In the subgroup analysis, adjusted and non-adjusted mortality risks in the patients with septic shock were more likely to increase due to failure or delay in achieving the hour-1 bundle compared to those in the patients without septic shock. However, the effect modification was not statistically significant (S1 Fig).

Among the components of the hour-1 bundle, non-adherence to obtaining blood cultures (OR 2.65 [95% CI 1.25–5.62]), administering broad spectrum antibiotics (OR 4.81 [95% CI 1.38–16.72]), and administering of crystalloid (OR 13.97 [95% CI 2.19–89.31]) within 1 hour were associated with increased in-hospital mortality among the patients with sepsis. In addition, the adjusted OR for 1-hour delay of achievement of the components of the hour-1 bundle

**Table 1. Patient characteristics.**

| | Non-bundle-adherent group | Bundle-adherent group | p Value |
|---|---|---|---|
| | n = 89 | n = 89 | |
| Age | 75 (67–84) | 75 (69–82) | 0.946 |
| Sex, male | 62 (70%) | 58 (65%) | 0.522 |
| BMI (kg/m$^2$) | 21 (18–24) | 22 (19–25) | 0.327 |
| Charlson comorbidity index | 1 (1–4) | 1 (0–2) | 0.073 |
| Site of infection | | | 0.092 |
| Lung | 42 (47%) | 40 (45%) | |
| Abdomen | 15 (17%) | 18 (20%) | |
| Urinary tract | 12 (13%) | 19 (21%) | |
| Bone soft tissue | 6 (7%) | 6 (7%) | |
| Cardiovascular | 0 (0%) | 2 (2%) | |
| Other unidentified | 14 (16%) | 4 (4%) | |
| Glasgow Coma Scale | 13 (10–14) | 13 (9–15) | 0.509 |
| Respiratory rate (/min) | 22 (19–30) | 20 (16–30) | 0.144 |
| Systolic blood pressure (mmHg) | 96 (74–119) | 92 (69–132) | 0.776 |
| Diastolic blood pressure (mmHg) | 50 (41–70) | 56 (42–72) | 0.371 |
| Mean blood pressure (mmHg) | 65 (53–86) | 69 (51–94) | 0.687 |
| Heart rate (bpm) | 105 (87–122) | 111 (91–129) | 0.259 |
| Body temperature (˚C) | 37.3 (36.3–38.5) | 38.2 (36.9–39.1) | 0.002 |
| Lactate (mmol/L) | 3.7 (2.1–5.6) | 3.7 (1.8–5.3) | 0.624 |
| White blood cell count (×10$^3$/μL) | 12.3 (7.7–17) | 9.7 (5.6–14.3) | 0.049 |
| Platelet count (×10$^4$/μL) | 18.3 (13.8–25.4) | 16.3 (11.8–22.5) | 0.137 |
| Total bilirubin (mg/dL) | 0.8 (0.6–1.2) | 0.9 (0.6–1.5) | 0.176 |
| Creatinine (mg/dL) | 2 (1–3.8) | 1.4 (1.1–2.6) | 0.178 |
| Glucose (mg/dL) | 141 (100–231) | 154 (108–200) | 0.976 |
| CRP (mg/dL) | 11.3 (5.3–22.2) | 10.6 (2.9–21.9) | 0.501 |
| 24-hour urine volume (mL) | 945 (430–1700) | 1220 (641–1703) | 0.279 |
| FDP (μg/mL) | 13.4 (8.1–32.6) | 12.3 (6.5–27.4) | 0.233 |
| D-dimer (μg/mL) | 5.8 (2.9–12.3) | 6.3 (2.4–11.6) | 0.439 |
| APACHE II score | 24 (18–28) | 24 (18–27) | 0.587 |
| SOFA score | 8 (6–10) | 8 (5–10) | 0.357 |
| Mechanical ventilation use | 26 (29%) | 44 (49%) | 0.006 |

*BMI* body mass index, *CRP* C-reactive protein, *FDP* fibrin/fibrinogen degradation products, *APACHE* Acute Physiology and Chronic Health Evaluation, *SOFA* Sequential Organ Failure Assessment.

for in-hospital mortality was 1.28 (95% CI 1.04–1.57) using a multivariable logistic regression analysis (Fig 3). The results were similar in a sensitivity analysis that excluded components with adherence rates that were too high (S2 Fig).

**Table 2. Primary and secondary outcomes in the two groups.**

| | Non-bundle-adherent group | Bundle-adherent group | p Value |
|---|---|---|---|
| | n = 89 | n = 89 | |
| In-hospital mortality | 27 (30.3%) | 16 (18.0%) | 0.054 |
| Ventilator-free days | 19 (0–28) | 21 (0–28) | 0.696 |
| Intensive care unit-free days | 15 (0–22) | 18 (0–23) | 0.24 |
| Length of hospitalization (days) | 19 (10–42) | 21 (10–46) | 0.827 |

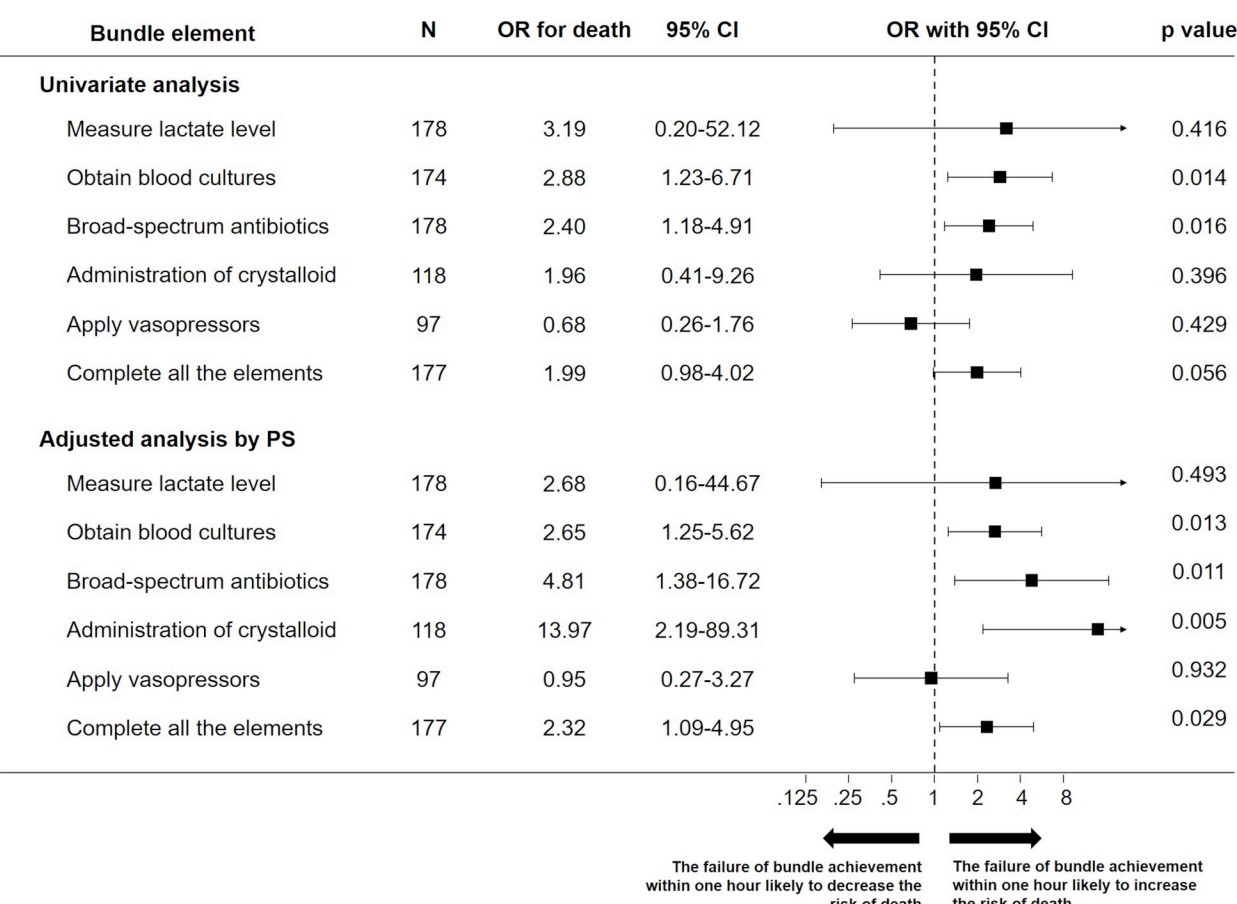

**Fig 2. Association between mortality and adherence to each bundle component within 1 hour.** Univariate and multivariate-adjusted ORs with 95% CIs for mortality risk are represented as forest plots. Inverse probability of treatment weighting analysis with propensity score was used as the adjustment method. *OR* odds ratio, *CI* confidence interval, *PS* propensity score.

## Discussion

Our study showed that approximately one-half of the patients with sepsis received care that adhered to the hour-1 bundle, and their outcomes were significantly improved when the hour-1 bundle was completed within 1 hour. However, among the hour-1 bundle components, only obtaining blood cultures and administering antibiotics may have contributed to the decreased in-hospital mortality.

Since the original studies of care bundle effectiveness were published [14, 15], there have been mainly negative opinions about its implementation [8, 16, 17]. In fact, our previous descriptive study did not prove the effectiveness of the 3-hour bundle [13]. However, few studies have directly investigated its effectiveness. Sepsis is different from other emergent conditions such as acute coronary syndrome, stroke, or trauma in terms of disease onset, although appropriate and timely detection and treatment should be essential in improving the outcome of sepsis. A recent retrospective cohort study showed no association between completion of the hour-1 bundle components within 1 hour and lower mortality, whereas bundle completion within 3 hours was associated with lower mortality [9]. In that study, the hour-1 bundle was completed in only 8% of the patients, whereas it was completed in half of the patients in the present study. Facilities with high bundle adherence rates may have better outcomes because of the multidisciplinary nature of the treatment. A better understanding and implementation

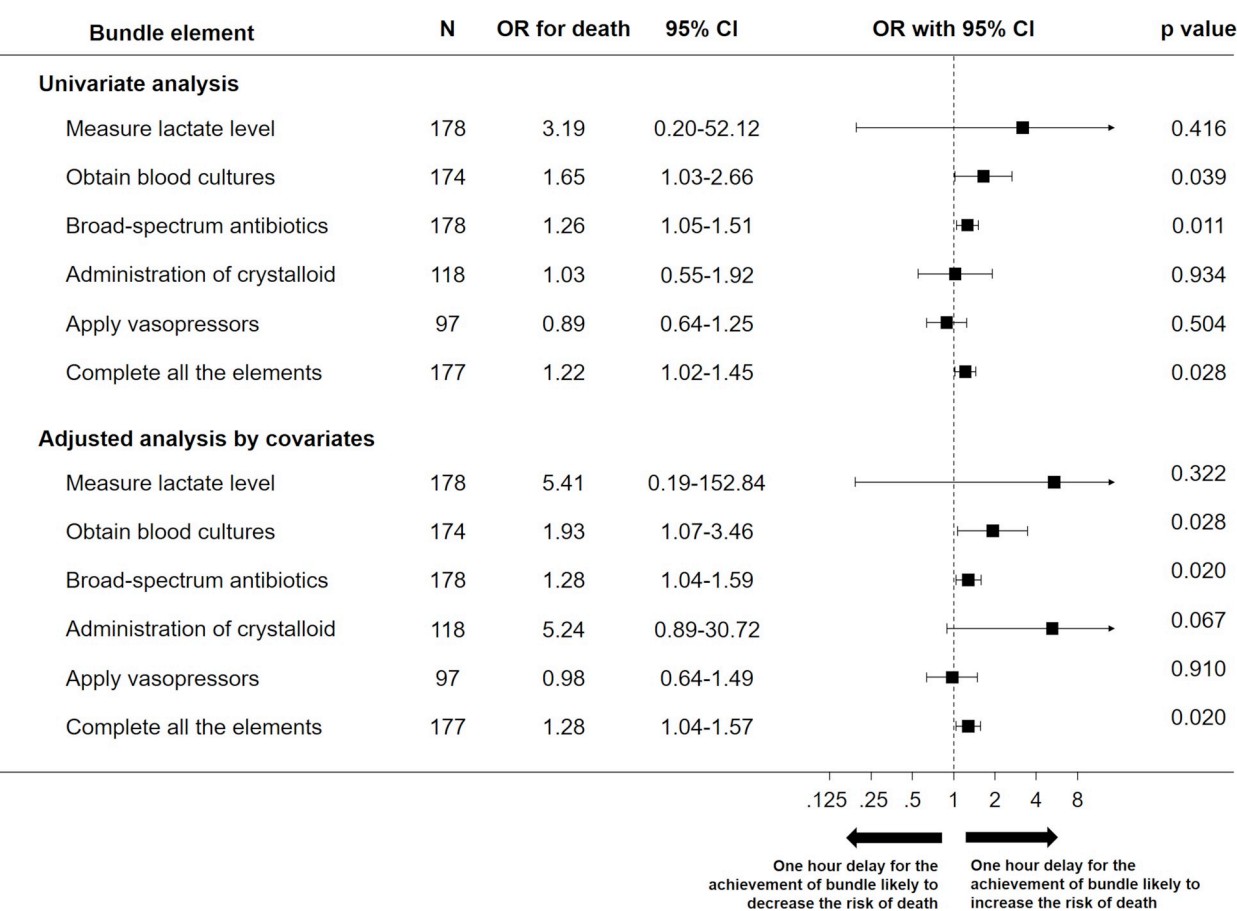

**Fig 3. Association between the increase in mortality and 1-hour delay in the achievement of each bundle component.** Univariate and multivariate-adjusted ORs with 95% CIs for mortality risk are represented as forest plots. Regression analyses were adjusted by including several clinically plausible and relevant confounders as covariates. *OR* odds ratio, *CI* confidence interval.

of the hour-1 bundle may be important for health care personnel to achieve an improvement in patient prognosis. For example, early goal-directed therapy (EGDT) has never been well recognized, and its effectiveness compared with usual care was only shown as it became better known. However, after most physicians came to understand it, the prognosis of sepsis treated with usual care based on EGDT knowledge and the SSC Guideline has been equal to or better than that for EGDT itself [18]. It is natural in health care research for results to change as education is disseminated. Thus, it may take some time until we see the effects of adherence to the hour-1 bundle.

Among the components of the hour-1 bundle, only obtaining blood cultures and administering antibiotics may have contributed to the decreased in-hospital mortality in the present study. These time-dependent factors were similarly significant whether they were dichotomized in 1 hour after diagnosis or as continuous variables every hour. Especially, early administration of antibiotics was adhered to in a considerably lower proportion of the study patients, and thus, this might strongly influence the study results. Early administration of antibiotics within 1 hour is sometimes difficult to achieve in a critical care setting. Actually, several previous studies reported similar or even lower proportions of patients receiving antibiotics within 1 hour [19, 20]. Multiple factors that might cause a delay in antibiotic administration include overcrowding, limited staffing, high patient load, blood culture testing, difficult intravenous line access, fluid resuscitation, and atypical presentation leading to delayed recognition of sepsis.

A systematic review of seven observational studies in the Japanese Clinical Practice Guidelines for Management of Sepsis and Septic Shock 2020 showed no significant difference in outcomes with administration of antibiotics within 1 hour versus sometime later [21]. Therefore, it was given a weak recommendation: initiation of antibiotics as early as possible, but not necessarily within 1 hour. However, we validated and agree with the importance of the hour-1 bundle as reported in previous studies [13, 14]. The study of a quality indicator such as time to antibiotic can influence the standardization of medical practice. However, education on sepsis care was not disseminated from previous studies that did not show an association with early antibiotic administration [17]. Actually, the results are more greatly influenced by strong clinical variables such as a certain treatment if clinically more important known or unknown (unmeasured) variables than a quality measurement such as time to antibiotic are assessed. It is most important that two groups with similar severity of illness receive similar treatments before looking at differences in time to antibiotic. Otherwise, patients receiving non-adherent care might experience worse outcomes because it may have been difficult to diagnose sepsis or its severity in these patients compared with patients receiving adherent care [22, 23]. The patients receiving non-adherent care had more unknown sources of infection, fewer fevers, and lower ventilator use. Thus, time to antibiotic might not have been a cause but a result of non-adherent care.

The resuscitation protocol of the hour-1 bundle such as maintenance of fluid volume and application of vasopressors will continue to be controversial. Adherence to the administration of crystalloid and lactate measurement were both very high in the present study. However, it is difficult to assess clinical effectiveness of a resuscitation protocol only from these results. Further study of fluid resuscitation and fluid balance is needed.

There are several potential approaches to reducing the delay in medical management and enhancing adherence to the hour-1 bundle in sepsis. First, greater education of hospital staff would improve their understanding and awareness of sepsis, leading to earlier diagnosis and treatment by first responders. Second, multidisciplinary collaboration that includes different health professionals could shorten the time to total medical contact and especially to antibiotic administration. As the optimal strategy to improve the quality of sepsis management might vary widely according to hospitals, regions, and countries, daily discussions among multidisciplinary professionals are important as a means of providing education to and enhancing the awareness of clinical staff participating in sepsis management.

Our study has several limitations. First, this study included convenience samples but not consecutive ones, which might have led to selection bias. Second, the adherence rate was very high. Regarding generalizability, as triage and the path to emergency care and intensive care are influenced by the health care system in each country, the current findings may not be applicable in different countries. Third, due to the nature of observational studies, the possibility remains that the prognosis was better for the patients who received bundle-adherent care within 1 hour than for those who could not receive it within this time. Fourth, the present study did not evaluate detailed reasons for the delay in or non-adherence to bundle care. Finally, the number of study patients was smaller than the pre-calculated sample size, which might have reduced the statistical power for detection of a true effect. Nevertheless, we reconfirmed that the time of antibiotic administration is a key component in the treatment of sepsis.

## Conclusions

We showed an association between completion of the hour-1 bundle and lower in-hospital mortality. Especially, administering antibiotics might have contributed the most to the

decrease of in-hospital mortality, and these findings would need to be confirmed in future further large-scale studies.

## Supporting information

**S1 Fig. Association between the increase in mortality and failure or delay in achieving the hour-1 bundle in subgroups with and without septic shock.** Univariate and multivariate-adjusted ORs with 95% CIs for mortality risk are represented as forest plots. Covariate adjustment or propensity score adjustment was used in the regression analyses as appropriate, and the significance level (p value) for effect modification was calculated. *OR* odds ratio, *CI* confidence interval, *PS* propensity score.
(DOCX)

**S2 Fig. Association between the increase in mortality and failure or delay in achieving three relatively low-compliance bundle components: Blood culture, antibiotics, and vasopressors.** Univariate and multivariate-adjusted ORs with 95% CIs for mortality risk are represented as forest plots. Covariate adjustment or propensity score adjustment was used in the regression analyses as appropriate. *OR* odds ratio, *CI* confidence interval, *PS* propensity score.
(DOCX)

**S1 Table. The 11 variables used to calculate propensity scores for adherence to the hour-1 bundle in the logistic regression models.** *SOFA* Sequential Organ Failure Assessment, *ICU* intensive care unit.
(DOCX)

**S2 Table. Time to completion of each component of the hour-1 bundle.** Missing data: Obtain blood cultures = 4; Administration of crystalloid = 1; Apply vasopressors = 2; completion of all elements = 1.
(DOCX)

**S3 Table. Time to completion of each component of the hour-1 bundle in the non-bundle-adherent group.**
(DOCX)

## Acknowledgments

We thank all of the JAAM MAESTRO study contributors and also Mr. Shuta Fukuda for his exceptional assistance in helping to complete this study.

## Author Contributions

**Conceptualization:** Yutaka Umemura, Toshikazu Abe, Hiroshi Ogura, Seitato Fujishima, Shigeki Kushimoto, Atsushi Shiraishi, Daizoh Saitoh, Toshihiko Mayumi, Yasuhiro Otomo, Kazuma Yamakawa, Taka-aki Nakada, Kohji Okamoto, Joji Kotani, Junichi Sasaki, Satoshi Gando.

**Data curation:** Yutaka Umemura, Toshikazu Abe, Seitato Fujishima, Atsushi Shiraishi, Toru Hifumi, Akiyoshi Hagiwara, Kazuma Yamakawa, Takehiko Tarui.

**Formal analysis:** Toshikazu Abe, Hiroshi Ogura, Daizoh Saitoh, Yasuhiro Otomo, Akiyoshi Hagiwara, Kazuma Yamakawa, Taka-aki Nakada, Takehiko Tarui.

**Investigation:** Yutaka Umemura, Toshikazu Abe, Hiroshi Ogura, Seitato Fujishima, Shigeki Kushimoto, Atsushi Shiraishi, Daizoh Saitoh, Yasuhiro Otomo, Toru Hifumi, Kazuma

Yamakawa, Yasukazu Shiino, Taka-aki Nakada, Takehiko Tarui, Joji Kotani, Yuichiro Sakamoto, Junichi Sasaki, Shin-ichiro Shiraishi, Ryosuke Tsuruta, Tomohiko Masuno, Naoshi Takeyama, Norio Yamashita, Hiroto Ikeda, Masashi Ueyama, Satoshi Gando.

**Methodology:** Yutaka Umemura, Toshikazu Abe, Hiroshi Ogura, Seitato Fujishima, Shigeki Kushimoto, Atsushi Shiraishi, Daizoh Saitoh, Toshihiko Mayumi, Yasuhiro Otomo, Toru Hifumi, Kiyotsugu Takuma, Yasukazu Shiino, Taka-aki Nakada, Kohji Okamoto, Joji Kotani, Yuichiro Sakamoto, Junichi Sasaki, Shin-ichiro Shiraishi, Ryosuke Tsuruta, Tomohiko Masuno, Naoshi Takeyama, Norio Yamashita, Hiroto Ikeda, Masashi Ueyama, Satoshi Gando.

**Project administration:** Yutaka Umemura, Hiroshi Ogura, Yasuhiro Otomo, Satoshi Gando.

**Validation:** Atsushi Shiraishi, Toru Hifumi, Akiyoshi Hagiwara, Kiyotsugu Takuma, Yasukazu Shiino, Takehiko Tarui, Kohji Okamoto, Yuichiro Sakamoto, Shin-ichiro Shiraishi, Ryosuke Tsuruta, Tomohiko Masuno, Naoshi Takeyama, Satoshi Gando.

**Writing – original draft:** Yutaka Umemura, Toshikazu Abe.

**Writing – review & editing:** Yutaka Umemura, Toshikazu Abe, Hiroshi Ogura, Seitato Fujishima, Shigeki Kushimoto, Atsushi Shiraishi, Daizoh Saitoh, Toshihiko Mayumi, Yasuhiro Otomo, Toru Hifumi, Akiyoshi Hagiwara, Kiyotsugu Takuma, Kazuma Yamakawa, Yasukazu Shiino, Taka-aki Nakada, Takehiko Tarui, Kohji Okamoto, Joji Kotani, Yuichiro Sakamoto, Junichi Sasaki, Shin-ichiro Shiraishi, Ryosuke Tsuruta, Tomohiko Masuno, Naoshi Takeyama, Norio Yamashita, Hiroto Ikeda, Masashi Ueyama, Satoshi Gando.

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
