## [Decision Letter · Decision Letter 0]

9 Nov 2021

PONE-D-21-31986Hour-1 bundle adherence was associated with reduction of in-hospital mortality among patients with sepsis in JapanPLOS ONE

Dear Dr. Abe,

Thank you for submitting your manuscript to PLOS ONE. After careful consideration, we feel that it has merit but does not fully meet PLOS ONE’s publication criteria as it currently stands. Therefore, we invite you to submit a revised version of the manuscript that addresses the points raised during the review process.

We look forward to receiving your revised manuscript.

Kind regards,

Ashham Mansur, MD, PhD

Academic Editor

PLOS ONE

Journal Requirements:

 [The funders had no role in study design, data collection and analysis, decision to publish, or preparation of the manuscript.] 

Additional Editor Comments:

The reviewers have raised major concerns regarding to the relatively small number of patients in relation to the number pf the involved centers. you need to adress this major issue adequately and make corrdctions to the conclusions respectively.

Reviewers' comments:

Reviewer's Responses to Questions

**Comments to the Author**

1. Is the manuscript technically sound, and do the data support the conclusions?

Reviewer #1: Partly

Reviewer #2: Partly

Reviewer #3: Yes

2. Has the statistical analysis been performed appropriately and rigorously? 

Reviewer #1: Yes

Reviewer #2: I Don't Know

Reviewer #3: I Don't Know

3. Have the authors made all data underlying the findings in their manuscript fully available?

Reviewer #1: No

Reviewer #2: No

Reviewer #3: Yes

4. Is the manuscript presented in an intelligible fashion and written in standard English?

Reviewer #1: Yes

Reviewer #2: Yes

Reviewer #3: Yes

5. Review Comments to the Author

Reviewer #1: Thank you for the opportunity to review this manuscript. This was a prospective, multi-center, observational study to validate the effectiveness of hour-1 bundle in Japanese patients with sepsis. They concluded that the adherence of hour-1 bundle was associated with decreased mortality risk, and the collection of blood culture and administration of antibiotics within 1 hour was independently associated with mortality risk in septic patients. I had several comments that the authors need to address:

1. The definition of hour-1 bundle adherence: the authors mentioned that "if a component of the bundle was not applicable, we treated achievement of the other components as completion of the bundle". I have concerns of that the authors mixed the non-shock and shock patients in this study (although the guideline did it so, and growing evidence suggested that these two groups should be treated differently). For example, 79 of 178 patients had no vasopressor administration (Table S1) and they were presumed to have low disease severity and mortality risk. What were the distribution of these non-shock patients in the adherence and non-adherence groups? They may significantly influence the study outcomes. I suggest the authors to perform subgroup analysis (shock: adherence vs.non-adherence, non-shock: adherence vs. non-adherence) in outcome analysis.

2. Since this study was conducted in multiple ICUs in Japanese hospitals, these patients were presumed to admit in ICUs,

but they had relative low proportion of vasopressor administration and mechanical ventilator use. They also had very

low CCI score.

3. Please clarify the origin of vital signs and laboratory results of each patient? from initial time of ICU admission? from

initial hospital admission? from the time of sepsis recognition?

4. It's surprising that the proportion of antibiotics administration within one hour was lower than vasopressor

administration (Table S1). Since the authors identifies the antibiotics administration as the independent factor associated with mortality risk, they may want to address more on this issue.

5. Similarly, the authors mentioned that they applied "multidisciplinary methods" to promote the adherence of hour-1

bundle in patients with sepsis. They should address more how they did it as the issue of quality improvement.

6. The recruitment period is different: in the design and setting "from July 2019 to August 2020." In the figure 1 "from July 2019 to December 2019" So the recruitment period is 13 or 6 months?

Reviewer #2: I read with interest the manuscript of Umemura et al. entitled “Hour-1 bundle adherence was associated with reduction of in-hospital mortality among patients with sepsis in Japan”.

In the present study the authors aimed to evaluate the effect of hour-1 bundle completion on clinical outcomes in sepsis. They showed an association between completion of the hour-1 bundle and lower in-hospital mortality in an observational multicenter study of 178 prospectively enrolled patients with sepsis.

The study’s conclusions are presented in an appropriate fashion and are supported by the data. The studies sample size of 178 individuals appears low in order to draw the presented conclusions, especially considering that 17 centers were involved into the study. The article is presented in an intelligible fashion and is written in standard English. However, the following comments/questions need to be addressed by the authors:

1. Introduction: “A recent study did not show an association between completion of the hour-1 bundle components within 1 hour and lower mortality, whereas they did show an association between care completed within 3 hours and lower mortality [9].”

Reference (9) refers to patients with septic shock and not patients with sepsis. The authors need to clarify that in the introduction or reference appropriate other studies.

2. The authors should provide a clear hypothesis at the end of the introduction based on clinical adjudication or existing research in the field.

3. How was the sample size of n=178 determined? The authors should provide a power calculation in order to verify that the sample size of n=178 is appropriate to show the study’s results at sufficient statistical power.

4. The authors state that 178 patients were enrolled into the study. 89 of the patients received bundle-adherent care (50%). However, it is stated in the results section that “the rate of completion of all components within 1 hour was 50.3%”. This is confusing. How do the authors explain 50.3% in table S1, while it should be 50%? Are the numbers correct?

5. The authors should clarify, if and how the performed multivariate analysis adjusts for the differences between the two groups at baseline (blood temperature and mechanical ventilation use).

6. “A systematic review of seven observational studies in the Japanese clinical practice guidelines for management of sepsis and septic shock 2020 showed no significant difference in outcomes with administration of antibiotic within 1 hour compared to later. “

The authors need to provide a reference for this in the discussion section.

Reviewer #3: Please get a native English speaker to correct the language in the manuscript. The lanugage is not up to the standard.

It might be reasonable to include the following in the limitation: The study did not anayse the reasons for non adherence.

It would be better if a graph/ table could show the time time period taken to complete the bundle in the non adherent group.

6. PLOS authors have the option to publish the peer review history of their article (what does this mean?). If published, this will include your full peer review and any attached files.

Reviewer #1: No

Reviewer #2: No

Reviewer #3: No

---

## [Author Response · Author response to Decision Letter 0]

3 Jan 2022

December 8, 2021

Ashham Mansur, MD, PhD

Dear Prof. Mansur

Manuscript Number: PONE-D-21-31986

Article Title: Hour-1 bundle adherence was associated with reduction of in-hospital mortality among patients with sepsis in Japan

We would like to thank all of the editors and reviewers for helping us improve our article. We revised our manuscript according to the comments and suggestions given by the reviewers. We also highlighted additional changes in the revised manuscript in red color with underline. Below are our point-by-point responses to the reviewers’ comments.

Additional Editor Comments:

The reviewers have raised major concerns regarding to the relatively small number of patients in relation to the number pf the involved centers. you need to adress this major issue adequately and make corrdctions to the conclusions respectively.

Reply: Thank you for your valuable suggestion. We have revised the entire manuscript to address the reviewers’ concerns regarding sample size. The conclusions were revised as indicated below;

Original sentence: “We showed an association between completion of the hour-1 bundle and lower in-hospital mortality. Among the components of the hour-1 bundle, obtaining blood cultures and administering antibiotics may have contributed the most to the decrease of in-hospital mortality.”

(Page 23, lines 3-6, in the CONCLUSIONS)

Revised sentence: “We showed an association between completion of the hour-1 bundle and lower in-hospital mortality. Especially, administering antibiotics might have contributed the most to the decrease of in-hospital mortality, and these findings would need to be confirmed in future further large-scale studies.”

Reviewer #1: Thank you for the opportunity to review this manuscript. This was a prospective, multi-center, observational study to validate the effectiveness of hour-1 bundle in Japanese patients with sepsis. They concluded that the adherence of hour-1 bundle was associated with decreased mortality risk, and the collection of blood culture and administration of antibiotics within 1 hour was independently associated with mortality risk in septic patients. I had several comments that the authors need to address:

1. The definition of hour-1 bundle adherence: the authors mentioned that "if a component of the bundle was not applicable, we treated achievement of the other components as completion of the bundle". I have concerns of that the authors mixed the non-shock and shock patients in this study (although the guideline did it so, and growing evidence suggested that these two groups should be treated differently). For example, 79 of 178 patients had no vasopressor administration (Table S1) and they were presumed to have low disease severity and mortality risk. What were the distribution of these non-shock patients in the adherence and non-adherence groups? They may significantly influence the study outcomes. I suggest the authors to perform subgroup analysis (shock: adherence vs.non-adherence, non-shock: adherence vs. non-adherence) in outcome analysis.

Reply: Thank you for your valuable comments and suggestions. We fully agree with your comment that the presence or absence of septic shock might influence the study results. Accordingly, we conducted a subgroup analysis and evaluated the associations between the increase in mortality and failure or delay in achieving hour-1 bundle in subgroups with and without septic shock. As a result, mortality in shock patients was more likely to increase by failure or delay in achieving the hour-1 bundle; however, effect modifications were not statistically significant. To address this information, we added a new supplemental Figure (Figure S2) and the following sentences.

(Page 13, lines 3-6, in the MATERIALS AND METHODS)

We added: “We also fitted logistic regression models to evaluate the association between the increase in mortality and failure or delay in achieving the hour-1 bundle in subgroups with and without septic shock, by including product terms between achieving the bundle and the presence of shock.”

(Page 17, lines 2-5, in the RESULTS)

We added: “In the subgroup analysis, adjusted and non-adjusted mortality risks in the patients with septic shock were more likely to increase due to failure or delay in achieving the hour-1 bundle compared to those in the patients without septic shock. However, the effect modification was not statistically significant (Fig S1 in the Supporting Information).”

Fig. S1. Association between the increase in mortality and failure or delay in achieving the hour-1 bundle in subgroups with and without septic shock. Univariate and multivariate-adjusted ORs with 95% CIs for mortality risk are represented as forest plots. Covariate adjustment or propensity score adjustment was used in the regression analyses as appropriate. The significance level (p value) for effect modification was calculated. OR odds ratio, CI confidence interval, PS propensity score

2. Since this study was conducted in multiple ICUs in Japanese hospitals, these patients were presumed to admit in ICUs, but they had relative low proportion of vasopressor administration and mechanical ventilator use. They also had very low CCI score.

Reply: Thank you for your valuable comments. As you mentioned, 44% of the study patients did not require vasopressor administration, and 61% did not require mechanical ventilation at the time of sepsis diagnosis. In Japan, septic patients diagnosed in emergency departments were typically admitted to ICUs, even when they did not require vasopressor administration or mechanical ventilation, to be monitored closely for progression of illness. Actually, a previous large-scale observational study* including septic patients in Japan a reported similar rate of vasopressor use. Also, 78.8% of the sepsis patients in the previous study had a CCI score of 2 points or less, which was equal to that of the present study. One possible reason for the low CCI score might be the lower prevalence of HIV/AIDS. The relatively higher rate of the population that receives a regular medical check-up in Japan might be another explanation for the low CCI score.

* Abe T, Ogura H, Shiraishi A, et al. Characteristics, management, and in-hospital mortality among patients with severe sepsis in intensive care units in Japan: the FORECAST study. Crit Care. 2018 Nov 22;22(1):322.

3. Please clarify the origin of vital signs and laboratory results of each patient? from initial time of ICU admission? From initial hospital admission? from the time of sepsis recognition?

Reply: Thank you for your valuable question. In the present study, we defined bundle initiation time as the time of sepsis recognition, and all presented vital signs and laboratory tests were recorded at that time. To address this point more clearly, we have revised the following sentence.

Original sentence: “Baseline characteristics and comorbidities were similar between the two groups, with the exception of body temperature and mechanical ventilation use (Table 1).”

(Page 14, lines 8-10, in the RESULTS)

Revised sentence: “Baseline characteristics, vital signs, laboratory test results, and severity scores obtained at the time of sepsis recognition were similar between the two groups, with the exception of body temperature, white blood cell count, and mechanical ventilation use (Table 1).”

4. It's surprising that the proportion of antibiotics administration within one hour was lower than vasopressor administration (Table S1). Since the authors identifies the antibiotics administration as the independent factor associated with mortality risk, they may want to address more on this issue.

Reply: Thank you for your valuable comments. We fully agree with your comment that the lower proportion (51.1%) of antibiotics administration within one hour in this study might be problematic. Unfortunately, however, several previous studies reported similar or even lower proportions of patients receiving antibiotics within 1 hour. A meta-analysis including 8 studies reported that only 3,335 (30.0%) among 11,017 patients with sepsis/septic shock received antibiotics within 1 hour after recognition of severe sepsis/shock. In the emergency department, there are several possible factors for the delay of antibiotics administration, including overcrowding, limited staffing, high patient load, blood culture testing, difficult intravenous line access, fluid resuscitation, and atypical presentation leading to delayed recognition of sepsis. We thus believe that early antibiotics administration, if successfully achieved, could decrease mortality risk in sepsis. To address these points more clearly, we have revised the manuscript and added two new references, as indicated below;

(Page 19, line 15 – page 20, line 6, in the DISCUSSION)

We added “Especially, early administration of antibiotics was adhered to in a considerably lower proportion of the study patients, and thus, this might strongly influence the study results. Early administration of antibiotics within 1 hour is sometimes difficult to achieve in a critical care setting. Actually, several previous studies reported similar or even lower proportions of patients receiving antibiotics within 1 hour [19, 20]. Multiple factors that might cause a delay in antibiotic administration include overcrowding, limited staffing, high patient load, blood culture testing, difficult intravenous line access, fluid resuscitation, and atypical presentation leading to delayed recognition of sepsis”

(Page 28, lines 7-12, in the References)

We added

“19. Sankar J, Garg M, Ghimire JJ, Sankar MJ, Lodha R, Kabra SK: Delayed administration of antibiotics beyond the first hour of recognition is associated with increased mortality rates in children with sepsis/severe sepsis and septic shock. J Pediatr 2021; 233:183–190.e3.

20. Sterling SA, Miller WR, Pryor J, Puskarich MA, Jones AE: The impact of timing of antibiotics on outcomes in severe sepsis and septic shock: a systematic review and meta-analysis. Crit Care Med 2015; 43:1907–1915.”

5. Similarly, the authors mentioned that they applied "multidisciplinary methods" to promote the adherence of hour-1 bundle in patients with sepsis. They should address more how they did it as the issue of quality improvement.

Reply: Thank you for your valuable comment and suggestion. Accordingly, we added several sentences to address how to improve the quality of sepsis management in regard to adherence to the hour-1 bundle and revised the manuscript as indicated below.

(Page 21, line 13 – page 22, line 5, in the DISCUSSION)

We added “There are several potential approaches to reducing the delay in medical management and enhancing adherence to the hour-1 bundle in sepsis. First, greater education of hospital staff would improve their understanding and awareness of sepsis, leading to earlier diagnosis and treatment by first responders. Second, multidisciplinary collaboration that includes different health professionals could shorten the time to total medical contact and especially to antibiotic administration. As the optimal strategy to improve the quality of sepsis management might vary widely according to hospitals, regions, and countries, daily discussions among multidisciplinary professionals are important as a means of providing education to and enhancing the awareness of clinical staff participating in sepsis management.”

6. The recruitment period is different: in the design and setting "from July 2019 to August 2020." In the figure 1 "from July 2019 to December 2019" So the recruitment period is 13 or 6 months?

Reply: Thank you for noticing this. “From July 2019 to August 2020” is the correct recruitment period in this study. We revised Figure 1 as indicated below.

Reviewer #2: I read with interest the manuscript of Umemura et al. entitled “Hour-1 bundle adherence was associated with reduction of in-hospital mortality among patients with sepsis in Japan”.

In the present study the authors aimed to evaluate the effect of hour-1 bundle completion on clinical outcomes in sepsis. They showed an association between completion of the hour-1 bundle and lower in-hospital mortality in an observational multicenter study of 178 prospectively enrolled patients with sepsis.

The study’s conclusions are presented in an appropriate fashion and are supported by the data. The studies sample size of 178 individuals appears low in order to draw the presented conclusions, especially considering that 17 centers were involved into the study. The article is presented in an intelligible fashion and is written in standard English. However, the following comments/questions need to be addressed by the authors:

1. Introduction: “A recent study did not show an association between completion of the hour-1 bundle components within 1 hour and lower mortality, whereas they did show an association between care completed within 3 hours and lower mortality [9].”

Reference (9) refers to patients with septic shock and not patients with sepsis. The authors need to clarify that in the introduction or reference appropriate other studies.

Reply: Thank you for your valuable comment and suggestion. We added a reference article to present more detailed evidence on the association between bundle adherence and mortality in both sepsis and septic shock and revised the manuscript as indicated below.

Original sentence: “A recent study did not show an association between completion of the hour-1 bundle components within 1 hour and lower mortality, whereas they did show an association between care completed within 3 hours and lower mortality [9].”

(Page 7, line 15 – Page 8, line 2, in the INTRODUCTION)

Revised sentence: “Several recent studies showed significant associations between bundle-adherent care completed within 3 hours and lower mortality. To date, however, little evidence has been provided of an association between completion of the hour-1 bundle components within 1 hour and lower mortality [9, 10].”

(Page 27, lines 1-3, in the References)

We added “10. Baghdadi JD, Brook RH, Uslan DZ, et al: Association of a care bundle for early sepsis management with mortality among patients with hospital-onset or community-onset sepsis. JAMA Intern Med 2020; 180:707–716.

2. The authors should provide a clear hypothesis at the end of the introduction based on clinical adjudication or existing research in the field.

Reply: Thank you for your valuable suggestion. Accordingly, we added the following sentence to clearly state the study hypothesis.

(Page 8, lines 4-7, in the INTRODUCTION)

We added: “Herein, we hypothesized that conventional bundle-adherent care within 3 hours would not be sufficiently effective to reduce the mortality of sepsis and septic shock, and adherence to the hour-1 bundle could play an important role in improving outcomes.”

3. How was the sample size of n=178 determined? The authors should provide a power calculation in order to verify that the sample size of n=178 is appropriate to show the study’s results at sufficient statistical power.

Reply: Thank you for your important question and suggestion. According to your suggestion, we described the detailed method to determine sample size in the present study. A total sample size of 300 patients was assumed to be necessary to conduct the pre-analysis plan. Unfortunately, however, we included a smaller number in the present study than the pre-calculated sample size. We thus addressed this point as a major limitation possibly reducing the statistical power in this study and revised the manuscript as indicated below;

(Page 11, line 12 – page 12, line 2, in the MATERIALS AND METHODS)

We added: “In total, 100 participants were required in the present study for the bundle-adherent group to conduct multivariate regression analyses. In a previous study conducted in Japan, early administration of antibiotics (within 1 hour) was adhered to in approximately 33% of septic patients [13]. Therefore, a total sample size of 300 patients was assumed to be necessary to include 100 patients with bundle-adherent care.”

(Page 22, lines 13-15, in the DISCUSSION)

We added: “Finally, the number of study patients was smaller than the pre-calculated sample size, which might have reduced the statistical power for detection of a true effect.”

4. The authors state that 178 patients were enrolled into the study. 89 of the patients received bundle-adherent care (50%). However, it is stated in the results section that “the rate of completion of all components within 1 hour was 50.3%”. This is confusing. How do the authors explain 50.3% in table S1, while it should be 50%? Are the numbers correct?

Reply: Thank you for noticing this. Actually, 50% is correct. We are sorry about this and have revised the wording to “50%”.

5. The authors should clarify, if and how the performed multivariate analysis adjusts for the differences between the two groups at baseline (blood temperature and mechanical ventilation use).

Reply: Thank you for your valuable suggestion. In the present study, adjusted regression analyses using propensity scoring for adherence to the hour-1 bundle were performed. The propensity score was calculated using multivariate logistic regression including 11 clinically plausible and relevant variables. Also, we performed a multivariable logistic regression analysis to estimate the risk of death associated with every 1-hour delay in achieving the bundle, adjusted by including the equal covariates. To address these points more clearly, we added a supplemental table (Table S1) describing the covariables used to adjust the regression analyses, and we revised the manuscript as indicated below.

Original sentence: “The impact of non-adherence to the hour-1 bundle on risk-adjusted in-hospital mortality was estimated using an inverse probability of treatment weighting analysis with a propensity score.”

(Page 12, lines 9-11, in the MATERIALS AND METHODS)

Revised sentence: “The impact of non-adherence to the hour-1 bundle on risk-adjusted in-hospital mortality was estimated by logistic regression analyses adjusted by an inverse probability of treatment weighting analysis using propensity scoring.”

(Page 12, lines 15-16, in the MATERIALS AND METHODS)

We added: “(Table S1 in the Supporting Information)”

6. “A systematic review of seven observational studies in the Japanese clinical practice guidelines for management of sepsis and septic shock 2020 showed no significant difference in outcomes with administration of antibiotic within 1 hour compared to later. “

The authors need to provide a reference for this in the discussion section.

Reply: Thank you for your suggestion. Accordingly, we added a new reference as Reference 21.

(Page 28, lines 13-14, in the References)

We added: “Egi M, Ogura H, Yatabe T, et al: The Japanese Clinical Practice Guidelines for Management of Sepsis and Septic Shock 2020 (J-SSCG 2020). Acute Med Surg 2021; 8:e659.”

Reviewer #3: Please get a native English speaker to correct the language in the manuscript. The lanugage is not up to the standard.

Reply: Thank you for your suggestion. We had our revised manuscript checked by an English language native speaker.

It might be reasonable to include the following in the limitation: The study did not anayse the reasons for non adherence.

Reply: Thank you for this valuable suggestion. We added the following sentence in the limitations paragraph.

(Page 22, lines 12-13, in the DISCUSSION)

We added: “Fourth, the present study did not evaluate detailed reasons for the delay in or non-adherence to bundle care.”

It would be better if a graph/ table could show the time time period taken to complete the bundle in the non adherent group.

Reply: Thank you for this valuable suggestion. Accordingly, we added the following new table as Table S3 in the supporting information to present the time period taken to achieve each component of bundle care in the non-adherent group.

---

## [Decision Letter · Decision Letter 1]

31 Jan 2022

Hour-1 bundle adherence was associated with reduction of in-hospital mortality among patients with sepsis in Japan

PONE-D-21-31986R1

Dear Dr. Abe,

We’re pleased to inform you that your manuscript has been judged scientifically suitable for publication and will be formally accepted for publication once it meets all outstanding technical requirements.

Kind regards,

Ashham Mansur, MD, PhD

Academic Editor

PLOS ONE

Additional Editor Comments (optional):

Reviewers' comments:

Reviewer's Responses to Questions

**Comments to the Author**

1. If the authors have adequately addressed your comments raised in a previous round of review and you feel that this manuscript is now acceptable for publication, you may indicate that here to bypass the “Comments to the Author” section, enter your conflict of interest statement in the “Confidential to Editor” section, and submit your "Accept" recommendation.

Reviewer #1: All comments have been addressed

Reviewer #2: All comments have been addressed

2. Is the manuscript technically sound, and do the data support the conclusions?

Reviewer #1: (No Response)

Reviewer #2: Yes

3. Has the statistical analysis been performed appropriately and rigorously? 

Reviewer #1: (No Response)

Reviewer #2: Yes

4. Have the authors made all data underlying the findings in their manuscript fully available?

Reviewer #1: (No Response)

Reviewer #2: No

5. Is the manuscript presented in an intelligible fashion and written in standard English?

Reviewer #1: (No Response)

Reviewer #2: Yes

6. Review Comments to the Author

Reviewer #1: (No Response)

Reviewer #2: Thank you for the provided point by point review. All mentioned concerns have been addressed sufficiently. I recommend to accept the manuscript in the current version.

7. PLOS authors have the option to publish the peer review history of their article (what does this mean?). If published, this will include your full peer review and any attached files.

Reviewer #1: No

Reviewer #2: No

---

## [Editor Report · Acceptance letter]

4 Feb 2022

PONE-D-21-31986R1 

Hour-1 bundle adherence was associated with reduction of in-hospital mortality among patients with sepsis in Japan 

Dear Dr. Abe:

I'm pleased to inform you that your manuscript has been deemed suitable for publication in PLOS ONE. Congratulations! Your manuscript is now with our production department. 

Kind regards, 

on behalf of

Dr. Ashham Mansur 

Academic Editor

PLOS ONE